# A Practicable Operationalisation of Meaningful Human Control

Jonathan Kwik

Faculty of Law, University of Amsterdam, 1012 WX Amsterdam, The Netherlands; h.c.j.kwik@uva.nl

**Abstract:** Meaningful Human Control (MHC) has been a consistent key term in legal debates concerning autonomous weapon systems (AWS), but its usefulness as a policy or lawmaking tool is limited due to a lack of clarity on what the concept encompasses. This study engaged in a thorough literature study of official statements, policy papers and academic papers published between 2013–2021 to determine features common to these proposals and synthesise a workable framework of MHC. The framework identifies five core elements—awareness, weaponeering, context control, prediction and accountability—and many interlocking mechanisms which link these elements together in a causal and chronological manner corresponding to the military targeting process. Subsequently, a detailed commentary and discussion is provided on the individual differences between sources, how specific elements can be implemented in practice by military commanders, and particularly controversial points are highlighted which require specific consideration by commentators and policymakers. The framework identifies concrete and practicable ways commanders can exercise control over AWS and serves as a solid foundation for further legal analysis of commanders' duties when employing AWS, for future policy discussions, and as a problem-solving tool to resolve important legal questions such as the ubiquitous 'accountability gap' conundrum.

**Keywords:** autonomous weapon system; meaningful human control; international humanitarian law; targeting; artificial intelligence; human-machine interaction; war; weapons; accountability gap

## 1. Introduction

After a decade since (Human Rights Watch 2012) published *Losing Humanity: The Case Against Killer Robots* in 2012, relatively little has actually been achieved in the formal sense concerning autonomous weapon systems (AWS). As militaries (Defense Science Board 2012; Knight 2019; Ministère des Armées (France) 2019) announced interest in harnessing artificial intelligence (AI) to allow weapon systems to make increasingly complex and independent decisions on the battlefield, the initial reverberations were significant: authors (Sparrow 2007, 2016; Asaro 2012) NGOs (Campaign to Stop Killer Robots 2012; Fleming 2009; Future of Life Institute 2015) and international organisations (Beerli 2014; Heyns 2013; ICRC 2013) swiftly called for discussions or even outright bans of the technology, and an intergovernmental panel was established under the auspices of the Convention on Certain Conventional Weapons (CCW) to discuss the appropriate international response (Simon-Michel 2014). More than eight years on, the output of the CCW panel has been poor, its discussions seemingly marred by semantics, a lack of focus, and conflicting political positions (Jensen 2020; Schuller 2017).[1]

One positive aspect of the CCW discussions is that it placed the notion of Meaningful Human Control (MHC) unto the international agenda. Initially advocated by NGOs, it rapidly gained traction and eventually became a central concept in not only CCW discussions but also literature, civil society and some official State positions (Eklund 2020; van den Boogaard and Roorda 2021). The notion of MHC is seen by many as a useful

---

1 For summaries of the CCW discussions, see (Group of Governmental Experts 2019; Marauhn 2018; Meier 2017).

foundation to base political and legal debates on (see Section 2.1), but like the AWS debate as a whole, suffers from substantive disorganisation. While almost all parties seem to agree that MHC is a desirable concept, individual positions differ significantly on what would constitute (or contribute to) achieving this MHC ideal. Stakeholders use different terms and advance different demands (ICRC 2016b), ultimately creating a of lack of consensus as to how it is to be operationalised or applied in practice (Jensen 2020). In essence, the international community has agreed on a goal, but struggles to determine the overarching mechanism to achieve it. With so many different positions proposed over the years, it has become hard to see the forest for the trees.

This paper builds upon our current understanding of MHC and advances it on the basis of two hypotheses. First, that the disparate and sometimes seemingly contradictory accounts of MHC, once sufficiently abstracted, in actuality form a coherent and consistent picture of the core elements necessary for exercising meaningful control. Second, that such elements do not exist independently but are contingent on the fulfilment of prior elements in the chronological chain. A careful analysis of existing expressions of MHC demonstrate these assumptions to be accurate and that such an integrated conception of MHC is indeed possible. The primary contribution of this paper, then, is a novel cognitive framework to comprehend and express the core requirements which underlie meaningful control. This cognitive framework can in turn help inform future academic debate and focus policy discussion around a uniform understanding of MHC and its constituent elements. In addition, we also will demonstrate how the framework can function as a powerful tool to establish accountability through a clear delineation of both knowledge and causality, which permits the allocation of responsibility for AWS-related violations.

The integrated framework as proposed in this paper draws from the past decade of discussions on MHC. The data used to build this framework is based on a thorough literature study of the different ways MHC has been conceptualised and how the notion has evolved over the years. Based on this, we construct a framework of *facets* and *processes*. Facets are nodes which represent categories such as 'awareness' and 'predictability': elements which in the international community's view allow us to achieve MHC. *Processes* represent the mechanisms which link aforementioned nodes into a chronological structure, providing not only causal representations of how facets interact with each other, but also placing the MHC framework into the practical domain of the military decision-making process.

This paper proceeds as follows. First, we briefly explore MHC has a notion, its history, its advantages and disadvantages as a guiding concept for AWS debates, its current shortcomings, and why the integrated framework proposed in this paper is beneficial to guide future discussions. We follow with a note on methodology, particularly how the data was collected, how facets and processes are differentiated, and how the framework fits within the military decision-making process. Then, we elaborate on each of the identified facets and processes in sequence, critically exploring the different ways these components are formulated. We end with some concluding remarks and recommendations for further research.

## 2. Background

### 2.1. Historical Overview

The first influential use of MHC as a concept can be traced to a 2013 paper by the NGO Article 36 (Article 36 2013a), published as a direct response to the rising controversy related to AWS. The term quickly gained popularity, and by 2014 was taken up by other organisations and States as a key concept to frame debates concerning autonomy in weapon systems (UNIDIR 2014). (UNIDIR 2014) and (Crootof 2016) both attribute its success to its intuitive appeal, providing some foundation to build discussions around despite its lack of clarity. By 2015, MHC had "emerged as a major theme" of many debates (Horowitz and Scharre 2015), including that of the CCW, and by 2016, (ICRC 2016c) reported how the "notion of human control has become the overarching issue in the debates on autonomous weapon systems".

By this time, many writers (ICRC 2016b; Moyes 2016; Roff and Moyes 2016) lauded MHC as a unifying force, acting as a point of coalescence between often-divergent political positions. (Roff 2016) optimistically wrote: "Whatever the terms ultimately become, there is consensus that no one wants weapons that operate out of human control." Critically, there was also State support that MHC should always be retained as more advanced AI are installed into weapon systems (Geiß and Lahmann 2017). Even now, MHC remains general common ground between both supporters and detractors of the technology alike (Boothby 2021; Ekelhof 2019; United Kingdom 2020).

Unfortunately, this apparent unity belies the fact that MHC as a concept is poorly established. When it was first introduced in 2013, MHC was still framed relatively nebulously. In fact, Article 36 noted that it was deliberately conceived this way. (Moyes 2016) explained: "[Q]uestions relating to what is required for human control to be "meaningful" are open. Given that openness, meaningful human control represents a space for discussion and negotiation." Over the following years of discussion, however, the international community never truly managed to fill the gaps left open by Article 36 to arrive at a coherent concept of what MHC exactly entails. This is not to say that stakeholders and commentators do not have their respective positions on the matter: there is no shortage of official statements, reports, and papers each advocating for what—in each respective source's view—constitutes MHC. What lacked was a unifying framework to merge these disparate views into a tangible and concrete set of requirements.

### 2.2. The Need for a Working Framework of MHC

Ever since MHC gained in popularity, there have been consistent calls (Boothby 2021; UNIDIR 2014) to make the notion more explicit and concrete, and for good reason. The disorganised nature of MHC in its current state makes it difficult to use to direct policy or legislation. (Santoni de Sio and van den Hoven 2018) note that "policy-makers and technical designers lack a detailed theory of what "meaningful human control" exactly means; and therefore they don't know which specific legal regulations and design guidelines should be derived from this principle." (Horowitz and Scharre 2015) are very critical of the continued use of MHC without first trying to consolidate these different positions and interpretations. Instead of presenting a solution, it risks becoming "only a pleasant-sounding catchphrase" or "an empty platitude, and one that would be devoid of a common meaning" at best, or leading to flawed policy choices at worst.

Without a clear blueprint of how to achieve MHC, it is difficult to make any informed policy decisions with it as a basis. This is true from both from a design perspective (such as that required during weapon development and acquisition) (Schuller 2017) and operational perspective (such as during targeting procedures) (Crootof 2016). It is also important from a legal standpoint. In international law, all belligerent parties are obliged to implement international humanitarian law (IHL), the law that applies to armed conflicts (Gill and Fleck 2010; Solis 2010). One core principle of IHL is that belligerents are limited in the choice of means of warfare they can implement (Additional Protocol I 1977), and this sets requirements for which weapons can be deployed against the enemy, and in what way (Fleck 2013). Therefore, if the international community wishes to use MHC to direct the legal debate—which it has frequently indicated[2]—crystalising MHC into a concrete framework is a paramount first step.

Why MHC can be useful as a legal tool has been highlighted by many authors. For example, (Geiß and Lahmann 2017) noted that it could be construed as an absolute legal requirement under IHL, the absence of which could render weapon systems illegal per se. Others (Article 36 2013b; Jensen 2018) have remarked that it can be used as a valuable standard for both pre-adoption reviews and post-deployment evaluations to ensure the capacity of weapon systems to comply with IHL requirements. Of course, using MHC in this way—as a proxy or shorthand for IHL requirements or as an indicator of a weapon's

---

[2] See e.g., (Article 36 2013b; Boardman and Butcher 2019; Greece 2019; ICRC 2016b).

legality—requires clarity on its component elements. Only once the scope of MHC is made clear can it be analysed in a legal sense, i.e., whether it (in whole or in part) constitutes a legal obligation under IHL, and on the basis of what specific provision or principle.

As an illustration, it has been argued that "[m]eaningful human control over the use of weapons is consistent with and promotes compliance with the principles of international humanitarian law, notably distinction and proportionality" (Human Rights Watch 2016) or that "human control over AI and machine-learning applications employed as means and methods of warfare is required to ensure compliance with the law" (ICRC 2019a). These statements are unfalsifiable in a legal sense until the exact contents of the 'human control' requirement are delineated first; as they are, they "add little substance to the discussion until States can either come to agreement or develop law through practice" (Jensen 2020). As it stands, MHC remains undefined—and thus unworkable—in international law (Chengeta 2016). The ICRC, aware of this problem, asked the international community to "consider the way in which humans must inject themselves into the decision-making process and at what points, to ensure this control is sufficient" (ICRC 2018). Let us thus turn our attention to creating a more concrete conception of MHC to aid both the policy and legal debate.

## 3. Methods

### 3.1. Finding Common Ground: A Demonstration

One common criticism of MHC as a concept is that it is vague and imprecise (van den Boogaard and Roorda 2021). The current author contends that the main reason for this lies in the fact that most elements are proposed as separate criteria without relating them with each other or placing them in the context of the military decision-making process. This results in loose, independent demands which sometimes overlap with the points advanced by different sources, but others which are mutually exclusive. In such a case, it becomes quite difficult to reach a consensus, especially when there are dozens of unique lists of requirements proposed by each stakeholder. As an illustration, let us take just the following two lists which each purport to describe key elements of MHC as seen in Table 1:

**Table 1.** Two sample lists of key MHC elements.

| Requirement | (A) Article 36 (Moyes 2016) | (B) (ICRC 2016b) |
|---|---|---|
| Requirement 1 | Predictable, reliable and transparent technology | Predictability of the weapon system |
| Requirement 2 | Accurate information on the outcome sought, the technology, and the context of use | Reliability of the weapon system |
| Requirement 3 | Timely human judgment and action | Human intervention during development, deployment and use |
| Requirement 4 | The potential for timely intervention | Knowledge about the functioning of the weapon system and the context of use |
| Requirement 5 | A framework of accountability | Accountability for its use |

From a cursory glance, it quickly becomes apparent that there are several points of overlap. However, even if we assume hypothetically that Article 36 and the ICRC were the only two stakeholders in existence to have proposed such a list, can we easily come to a unified conception of MHC that comports to both? It is quite challenging. It may require some re-arranging or sub-dividing of the points, and even then, there will be exclusive demands not present in the other set (e.g., 'transparent technology' is not present in the ICRC set). In truth, Article 36 and the ICRC align quite well in their respective formulations of MHC elements, and there are much more divergent positions out there. For example, (Marauhn 2018) argues that MHC is fundamentally a question of guaranteeing cognition ("knowledge of the decision-maker about the consequences of particular decisions and acts") and volition ("intent follows from human involvement and remains there, either

through supervision, programming or deployment"). It is evident that continuing this process with dozens more of such lists is unworkable, and possibly also can explain why it has been so difficult to arrive at a reasonable consensus in the international debate.[3]

The current author, then, argues that it is unhelpful to focus on isolated criteria to conceptualise MHC, as it tends to overemphasise differences between sources. Instead, we can try to focus on common elements. To return to the example of Table 1, what are the major themes that can be distilled from both lists? The first one seems to be awareness of the system's underlying technology and its working environment (A2, B4). To extend this point, Marauhn's 'cognition' element follows naturally from this: an understanding of how a system works and how it will be used will also allow the decision-maker to understand the consequences of such use. The cognition element also seems to imply some form of prediction, which aligns well with A1 and B1. In contrast, Marauhn's 'volition' element emphasises human involvement throughout the broader weapon life-cycle. It agrees well with B4 and partially with A3 and A4 (which, as they are formulated, are limited to actual weapon use). Finally, A5 and B5 refer to accountability, which Marauhn's conception lacks. As we are working based on commonalities instead of discrepancies, however, let us ignore this for the moment.

This methodology preliminarily yields the following greater elements: (1) understanding/awareness; (2) some involvement in the life-cycle; (3) prediction; (4) accountability. Of course, each of these major elements contain smaller (and possibly conflicting) operationalisations, but this rough collection of four points is already much wieldier as a working conception than dozens of individual sets. Furthermore, we have noticed something interesting, namely that there is a form of entanglement between the elements. We have seen this in the fact that awareness of the system and its context of use leads to another proposed criterion: predictability. Such causal mechanisms connect the isolated MHC elements, as they explain how the elements are intertwined and influence each other.

This short sub-section has provided an illustration of the thought process that initially formed the foundation of the framework proposed below (see Section 4). Before moving to the framework itself, it is necessary to briefly explain how the data which underpins the framework was collected.

### 3.2. Methodological Process

In Section 3.1, we applied a deductive process on the basis of three sample lists to arrive at our four generalised elements. A similar process, on a broader scale, was performed by Eklund in a 2020 report titled *Meaningful Human Control of Autonomous Weapon Systems: Definitions and Key Elements in the Light of International Humanitarian Law and International Human Rights Law* (Eklund 2020). In that study, Eklund collected definitions of MHC from NGOs, international organisations and States, and identified recurring similarities and differences in those definitions from an IHL perspective. The data subsequently yielded six "key elements of MHC".[4]

---

3  Another feasible explanation, posited by (Chengeta 2016), is that there are irreconcilable normative views between stakeholders and that, as a result, they (either deliberately or not) will indefinitely postpone agreeing on the criteria for MHC.

4  Eklund's elements can be summarised as follows:

- *Context control*: Controlling the context in which an AWS will operate to minimise the inherent uncertainty attached to using a weapon system in a dynamic environment.
- *Understanding the system*: Awareness of the system's abilities, limitations, and how it reacts to the environment.
- *Understanding the environment*: Awareness of the operational environment in which the system will operate, to enable proper adaptation to changing situations.
- *Predictability and reliability*: Ensuring that the system works in a way that it corresponds with the deployer's intent.
- *Human supervision and ability to intervene*: That humans can receive information from the system, and can relay information or commands back to it.
- *Accountability*: The ability to assign responsibility to humans for unlawful acts caused by the system.

Eklund's study was a significant step forward in systematising the wealth of different positions, and provides a solid foundation for further developing a unified conception of MHC. In addition, similarly to what we had experienced ourselves in Section 3.1, Eklund also captured glimpses of underlying mechanisms tying different key elements together. For example, in the section on predictability and reliability, she remarks that

> by using "context-control" the predictability increase, by "understanding the weapon" the use of it will be more predictable and by "understanding the environment" the autonomous weapon system's interaction with it will be more predictable.

Eklund, unfortunately, did not pursue this observation further in the report. As a result, her six key elements are pertinent, but remain relatively disjunctive criteria.

The current paper builds upon what Eklund has achieved in two ways. First, we attempt to reorganise and reformulate the six elements into clearly defined *facets*: criteria, defined as concretely as possible, in the form of requirements that decision-makers must implement to attain MHC. The facets are represented as nodes in our framework, which allows for further division into sub-nodes if necessary. Second, we expand upon the observation that facets may be interlinked by also considering *processes*. These are the mechanisms linking the nodes representing causal relationships. For example, using this method, Eklund's observation cited in the block quote above can be drawn as in Figure 1:

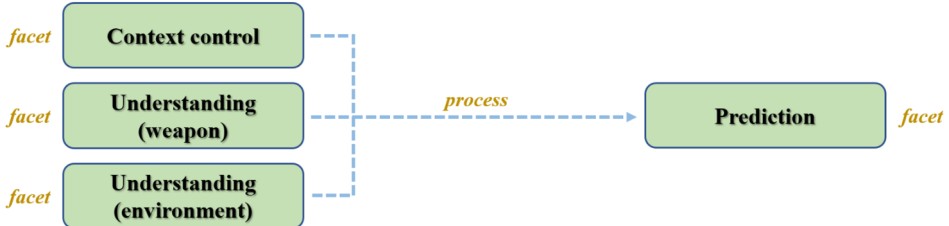

**Figure 1.** Sample diagram of facets and processes based on Eklund's observation.

The processes also inject a chronological element into the framework. Because mechanisms imply cause-and-effect, we can determine which facets come earlier in the military decision-making process and which become relevant later, perhaps during different operational phases. By combining facets and processes, we not only develop more concrete formulations of MHC elements, but also generate a clear sequence for decision-makers to follow when utilising these weapons in practice.

### 3.3. Data Collection and Processing

Like in Eklund's study, the data used to develop the framework proposed in this paper was drawn from a broad study of statements, reports and literature from NGOs, international organisations, States, and individual authors, from the period of 2013–2021. To be eligible, the statements had to include some claim, either directly or implicitly, that the proposed condition would 'contribute to', be 'necessary for', be a 'precondition for', or be a 'key element' for attaining MHC. Statements not purporting to be exhaustive lists, e.g., those only arguing for or explaining one particular element, were also included. All statements were given equal weight irrespective of source.[5]

After the data was collected, the six categories identified by Eklund were used as a foundation for preliminary coding, classifying each element qualitatively into one of the six categories. As more data points became available, the major categories were adjusted

---

A seventh point, 'Ethical considerations and the principle of human dignity', is also given as a key element, but is formulated more as a legal/moral *basis* for MHC rather than an expression of MHC itself. As such, it is not included in the list presented above.

[5]　As not all sources use the complete term 'meaningful human control', the terms 'human control' or 'control' were deemed sufficient for inclusion, as long as it was clear that the paper/statement was written in the context of the MHC debate or that the author intended it to be read in that context.

for precision either by fusing two categories into a broader one, or dividing the categories into sub-facets which more accurately represented the different criteria. This resulted in five major facets:

- [*AW*] Awareness
- [*WP*] Weaponeering
- [*CC*] Context Control
- [*PR*] Predictability
- [*AC*] Accountability

In this paper, facets are indicated by italic capitals inside square brackets. The first two facets also feature more detailed sub-facets, noted with subscript (e.g., [$AW_T$]) which are explored in more detail below.

The collected data was also analysed separately for processes. As processes refer to causal mechanisms linking different facets together, the key words used to identify and label them reflected this. These include phrases such as 'provides', 'is a function of', 'is linked to', 'would require', 'can be achieved by', and 'will affect'. An expansive substructure of interlinking mechanisms was found which connects all facets and sub-facets in robust ways, displaying a clear chronological forward process consistent with the overall practice of military decision-making. To illustrate this, the facets were associated with integers 1–6 and sorted accordingly, reflecting their order of application (see Section 4).

Section 4 shows the overall framework obtained from this analysis and some general remarks. In Sections 5 and 6, subsequently, we discuss each element of the framework in greater detail. As sources used in the construction of this framework frequently contribute to several components at once, the decision was made for the sake of clarity to reference these sources in the more applied Sections 5 and 6 that follow, as this allows linking them more directly to the specific elements of the framework they have contributed toward. Readers wishing to verify the datapoints used in the construction of the framework shown in Section 4 are thus encouraged to directly consult Sections 5 and 6, where these sources are cited extensively.

## 4. Results: The Integrated Framework

The proposed framework of MHC, built on the basis of an overall analysis of the data acquired, is shown in Figure 2.

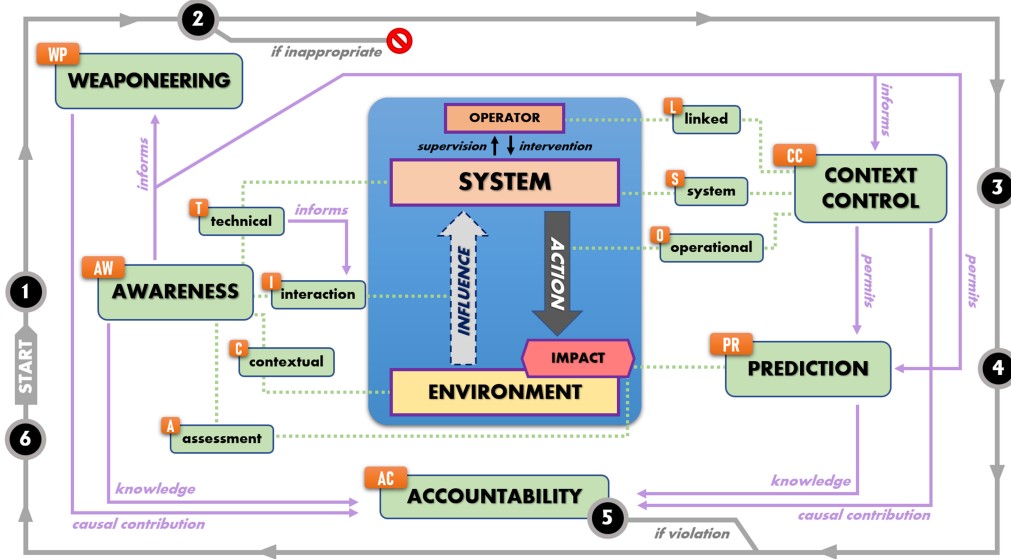

**Figure 2.** Integrated MHC framework.

The framework was conceptualised around the two central elements which constitute any weapon use: the weapon system itself ('*System*') and the operational environment

('*Environment*') in which it is used. This is illustrated by the blue central box. The AWS and its environment influence each other in both directions. (Frank et al. 2001) summarise this mutualistic relationship well: environments "provide percepts to the agent" and "the agent performs actions in them".[6] The environment determines what we sense, and thereby determines how we act upon it: this is represented by the *Influence* arrow in the centre. Simultaneously, decisions made by the agent have tangible effects on, and alter, the environment. This is represented by the *Action* arrow, which produces an *Impact* in the *Environment*. Ideally, from a military perspective, this impact would overlap with the desired effects of that operation, or the commander's "desired end state" (Curtis E. Lemay Center 2019), although this is not necessarily the case.

*Facets and sub-facets* in this framework are represented by green boxes with their respective abbreviations in the top-left corners. For brevity, these abbreviations will be used in subsequent sections. For instance, 'contextual awareness' will be noted as $[AW_C]$. The purple arrows connecting the facets represent the *processes*. The use of arrows instead of lines reflects the fact that processes are directional: they represent cause-and-effect. Processes are noted by a right arrow connecting two (sub-)facets, e.g., $[AW_C \rightarrow PR]$.

As an AWS is deployed in the context of a broader military operation, the most important actor of this process is the Deployer: they are the main protagonist of this framework. While this is not a military term per se, it was chosen for its descriptive value: The Deployer in this scheme refers to the person who ultimately makes the decision to deploy the AWS in the first place. In standard military processes, this usually will refer to the person(s) authorised to determine capability assignment, i.e., weapon selection and tuning (Curtis E. Lemay Center 2019; North Atlantic Treaty Organisation 2016), but it can flexibly refer to any officer who makes a similar decision. The Deployer is not featured[7] in this framework as they oversee the entire cognitive process: indeed, it is precisely the Deployer's thoughts and cognitive decision-making process that we map through this framework.

The decision to place the Deployer at the centre of this process reflects both literature and military practice. There is general agreement in policy and literature that the Deployer is the party primarily responsible for a weapon's usage and should be made the focal point of any MHC scheme. The 2019 NATO Allied Joint Doctrine (North Atlantic Treaty Organisation 2019) states this in no uncertain terms: "The commander is ultimately responsible for accepting risk." Similarly, (United States of America 2017) finds that "[i]n all cases, the commander is accountable and has the responsibility for authorizing weapon release in accordance with IHL". Several authors have also argued that assigning overall responsibility to the deployer has a clear legal basis in IHL, drawing from either the obligation to Respect and Ensure Respect (Stürchler and Siegrist 2017), the limitation on choosing means and methods of warfare (Bothe et al. 2013), or targeting principles requiring deployers to exercise precautionary measures over the effects of their deployed weapons (van den Boogaard and Roorda 2021).

As was established through the discovery of processes that link different facets together, there exists a causal mechanism that implies a chronological forward flow: the facets cannot be fulfilled out of sequence. This is captured by the outermost grey-coloured loop. The process *STARTs* at the step indicated by the number 1 and continues cyclically. This agrees with the iterative nature of the military targeting cycle (Huffman 2012; North Atlantic Treaty Organisation 2019). Occasionally this cycle may conclude early, as shown after Step 2 with the potential decision to not deploy a particular weapon, or branch off into additional obligations, such as finding accountability in cases of violations (Step 5).

---

[6]　In information technology, a percept refers to any input from the environment sensed by the agent, which it subsequently uses to make decisions (Winikoff et al. 2001).

[7]　Another actor *is* featured in the graph, however: the *Operator*. Their role is tied to facet $[CC_L]$ and limited to supervision and intervention. This is explained in more detail in Section 5.3. Under this framework, the primary cognitive burden, and thus responsibility for the deployment decision, remains with the Deployer.

Finally, the decision to *START* at Awareness (Step 1) instead of the actual weapon assignment (Step 2) also is consistent with the law and military practice. A commander's duties under IHL do not start with weapon selection, but include any prior due diligence necessary to make this decision. Deployment decisions are always preceded by extensive prior procedures of goal setting, target analysis, the gathering of intelligence, surveillance and reconnaissance (ISR) and weapon analysis (Ekelhof 2016; Roorda 2015). Throughout these steps, compliance to IHL requirements such as distinction and proportionality is rigorously tested (Ducheine and Gill 2018). Indeed, depending on how far we wish to expand the notion of the military decision-making process, preparatory decisions leading up to the weapon's adoption (acquisition policy, design, review and testing) can also be seen as contributing toward the ultimate deployment decision (Boardman and Butcher 2019). On this basis, some (Boothby 2021; de Jonogh 2019) have suggested that the concept of MHC should also include these prior phases. The current framework does not go as far, and limits itself to the operational phase.[8] The main point of this paragraph is to illustrate that the decision to deploy should be viewed more as a culmination of many prior considerations. Prior to this ultimate decision, the entire [*AW*] requirement must be completed first, which 'feeds into' the Deployer's decisions in [*WP*] and [*CC*].

Having briefly explained the overall structure of the framework, we now elaborate on each individual facet and process in turn.

## 5. Facets

### 5.1. [AW] Awareness

"To command effectively, the human operator must be involved . . . To be involved, the human operator must be informed." This excerpt from a NASA design principles report (Billings 1991) captures the Awareness facet in its purest form. The base philosophy is that "[h]uman control in the use of a technology is . . . based upon those planning and deciding upon an attack having certain information." (Article 36 2016) The ultimate goal of the Awareness facet is to empower the Deployer with the ability to make informed, conscious choices about the use of a weapon (Boardman and Butcher 2019; Horowitz and Scharre 2015).

The requirement of Awareness is usually expressed as encompassing a comprehensive understanding of the system-environment-interaction triad that lies in the centre of our framework. (UNIDIR 2014) asserts:

> Control is first and foremost based on knowledge of the weapon system. A thorough understanding by the operator and the commander of the selected weapon's functions and effects, coupled with contextual information (such as awareness of the situation on the ground, what other objects are in the target area, etc), contribute to the assessment of whether a weapon is appropriate for a particular attack.

According to (ICRC 2016c), MHC requires "[k]nowledge and accurate information about the functioning of the weapon system and the context of its intended or expected use". In the views of (van den Boogaard and Roorda 2021), any decision to deploy should be informed by "the capacity to understand the functioning of a system and the environment in which it will operate [and] the capacity to understand the possible effects that a system will generate considering the system's functioning and the environment". This last citation captures the totality of the system-environment-interaction triad in sequence.

As such, in the framework, Awareness is further developed into 3 sub-facets: [*AW_T*] Technical awareness, [*AW_C*] Contextual Awareness and [*AW_I*] Interaction Awareness.

**[*AW_T*] *Technical Awareness*** refers to an understanding of the system which is being considered for deployment. This sub-facet is summarised well by (Santoni de Sio and van den Hoven 2018): "In order for the system to remain under meaningful human control

---

8    This does not detract from the fact that these sources convincingly argue that human judgment is also exercised, often in decisive ways, prior to the targeting process.

we need thus to ensure that military commanders have a sound understanding of the function, capabilities, and limitations of the autonomous weapon technologies available to them." (ICRC 2016b) agrees that "[h]uman control on the part of the commander is based on sufficient knowledge and understanding of the weapon's functioning and proper training to ensure that, when deployed in a specific situation, it operates in accordance with IHL and any other restrictions that apply to its use".

Obtaining a proper understanding the system's capabilities and limitations seems to be a dominant bottom line (ICRC 2019b). (United States of America 2017) emphasised that the Deployer must be aware of the "system performance, informed by extensive weapons testing as well as operational experience". Naturally, this encompasses both the physical characteristics of the system (e.g., the type of munitions, sensors, chassis, method of attack) (ICRC 2016b; Schuller 2017) as well as its software component (how it makes decisions, what tasks it performs, its accuracy metrics) (Moyes 2016; United States of America 2017). An understanding of the system's reliability and robustness is frequently mentioned as necessary for making informed decisions (Geiß and Lahmann 2017; ICRC 2018; Moyes 2016). For instance, (Article 36 2017) raised the danger of Deployers not realising that mismatched proxy variables may lead to incorrect decisions. A famous example from the AI field is that of a neutral network trained to classify enemy and friendly tanks and which apparently had a high accuracy rate. Upon later inspection, however, it was found that the AI classified friendly from enemy tanks based on whether the sky was overcast or sunny (Freitas 2014).

In this context, it has also been observed that Technical Awareness may require some preparatory steps to be implemented, i.e., ranging beyond just the targeting cycle. For one, States may have to design for some degree of AI transparency (Boardman and Butcher 2019; ICRC 2018; Moyes 2016). This term, which we had encountered previously in Table 1, refers to the condition where a user understands why an AI makes the decisions it does (Miller 2019). In many modern AI, this is no longer necessarily the case due to the proliferation of machine learning techniques (Molnar 2019). While there are workarounds for this problem (Arya et al. 2019), these have to be implemented during design and not when the Deployer is at the point of deploying the system. In addition, even with these techniques, overcoming the 'black box' problem in modern AI remains quite challenging (Holland Michel 2020). (Kwik and Van Engers 2021) have argued that insufficient [$AW_T$] due to a lack of transparency would, in itself, preclude any IHL-compliant use, and that the Deployer must take this into consideration when deciding to field or reject the system. Another—more evident and banal—preparatory step for Technical Awareness is properly giving technical training to the Deployer (Geiß and Lahmann 2017; Horowitz and Scharre 2015; ICRC 2016b). "Training with systems allows users to understand and predict system behaviours across different situations in order to avoid undesirable outcomes or failures" (Boardman and Butcher 2019).

*[$AW_C$] Contextual Awareness* refers to an understanding of the environment in which the system will operate. In most sources, this is referred to as awareness of the 'context of use' (ICRC 2016c; Moyes 2016). The main rationale is that a "human commander cannot meet her obligations if she lacks sufficient information about context" (Roff 2016). This sub-facet is already very well understood by militaries and is readily implemented through the emphasis on ISR prior to any operation (Curtis E. Lemay Center 2019; Roorda 2015). As systems may only be specifically designed to work in particular conditions,[9] it is key that the Deployer verifies that the environment matches the system's design limitations.[10] Sources (Chengeta 2016; ICRC 2018) also emphasise the importance of continuously updating one's Contextual Awareness. As conditions on battlefields can change rapidly, it is crucial that decisions be made with the most up-to-date information (Roff and Moyes 2016). During

---

[9] This is often referred to as 'narrow AI' (Scharre 2014).
[10] For example, if an AWS is not designed to make classifications between civilian persons and combatants, it is necessary for the Deployer to know if civilians will be present in the field (Geiß and Lahmann 2017).

targeting procedures, this is often effectuated through requesting frequent updates or ISR validation leading up to the deployment decision (Roorda 2015).

*[AW$_I$] Interaction Awareness* is a more advanced form of Technical Awareness where not only the system itself is understood, but also how it would react to external stimuli. A weapon will react in certain ways to the nature of the environment or from direct interaction with it (ICRC 2018). Interaction Awareness is an essential component of control for all weapons: an artillery officer who is unable to account for the effect of wind on the ballistic trajectory of his mortars is a bad commander. As such, "military commanders and staff should be familiar, to a reasonable standard, with how a particular weapon would respond to potential situations" (Roorda 2015).

Interaction Awareness also encompasses awareness of the effects of potential countermeasures enacted by the enemy. Countermeasures are an inseparable part of warfare: "Every revolutionary, offensive technology is eventually met with a defensive counter" (Puckett 2004). In the case of AWS, the enemy will use both primitive and advanced techniques to degrade and interfere with the system's proper functioning, and perhaps even take over control of the system altogether (Boothby 2019). The precautionary principle in IHL requires attackers to take account of foreseeable enemy countermeasures which could impact the civilian population (Estreicher 2011). As such, to ultimately exercise and retain control, Deployers must be "aware of the operational effect, if any, of such intrusion or interference" (Boothby 2021).

The Awareness facet is also an effective tool to combat a critique levied against the use of supervisory control structures. The critique in question argues that even if a human (the 'controller') ostensibly remains in control of the weapon's operations (e.g., by maintaining what is often referred to as a human-on-the-loop[11]), they may not necessarily be in *de facto* control. "[I]ncluding a human in the lethal decision process is a necessary, but not a sufficient requirement" (Asaro 2012). Many examples are provided in literature illustrating how without general Awareness, being able to technically 'control' a weapon system amounts to nothing. Several reasons for this are proposed:

- *Ignorance of the system.* The controller is poorly trained (Asaro 2012), has an incomplete understanding of the system (Cummings 2004), does not understand how it makes decisions (UNIDIR 2016), or insufficiently appreciates its capabilities (Santoni de Sio and van den Hoven 2018).
- *Automation bias.* The controller places too much trust in the AI which leads to complacency (Chengeta 2016; Leveringhaus 2016), while in reality, they are unjustifiably overestimating (or ignorant of) the AI's accuracy or reliability rates (Parasuraman et al. 2000).
- *Lack of situational awareness.* The controller does not have sufficient context of the environment in which the system is operating and how it can impact the system to make a correct intervention (Cummings 2004; ICRC 2018).

In all these cases, sources argue that there is a lack of meaningful human control. This illustrates that even in situations which ostensibly involve a human directly in the decision-making process, the principles contained in the Awareness facet can be applied to evaluate whether MHC is retained. Naturally, a Deployer can fall into the same traps (e.g., making erroneous decisions based on technical or situational ignorance, as we have explored previously). It is the Awareness facet which remedies this issue.

### 5.2. [WP] Weaponeering

Having considered all relevant information from [AW], the Deployer now engages in the most powerful form of control they have over any possible type of weapon: the decision of whether or not to use it in the first place. Weaponeering is the "process of determining the specific means required to create a desired effect on a given target" (Office of the Chairman of the Joint Chiefs of Staff 2020), and at this point, the Deployer can

---

11    This refers to a human always being able to veto a system's decisions (Scharre and Horowitz 2015).

opt out from using an AWS if they conclude it is unable to fulfil IHL requirements or is unfit to achieve the mission goals (Moyes 2016). This is illustrated in Figure 2 through the branching path from Step 2 leading to a rejection of the AWS if deemed inappropriate for the overall circumstances identified in [*AW*]. [*WP*] is a straightforward facet but its importance cannot be overstated. Weaponeering constitutes the ultimate form of human control. "The military commander's decision to employ a certain a weapon in a certain context is the ultimate failsafe; a lack of capacity to understand the possible effects of the use of a system, may be fully compensated by a full capacity to decide on its use" (van den Boogaard and Roorda 2021).

*5.3. [CC] Context Control*

Once a Deployer chooses to assign an AWS, they can exercise advanced forms of Context Control to enforce additional constraints on the mission, described by (United States of America 2017) as the "employment of tactics, techniques, and procedures for that weapon". (United Kingdom 2020) elaborates that Context Control can take the form of "restricting the type of target and task, temporal and spatial constraints, constraining weapon effects, allowing for deactivation and fail-safe mechanisms where appropriate, and controlling the environment to exclude civilians or civilian objects". From this, we can divide Context Control into the following sub-facets: [$CC_S$] System Control, [$CC_O$] Operational Control, and [$CC_L$] Linked Control.

**[$CC_S$] *System Control*** refers to 'tweaking the system's parameters'. The Deployer can restrict the tasks that the system will perform, impose limits on the targets it may engage, the way it engages such targets, and, in the case where the platform carries more than one type of munition, prohibit the use of certain ones (ICRC 2019b; United Kingdom 2020). The Deployer can also consider whether they wish to enable online learning. This refers to whether or not the AI is allowed to continue incorporating experiences to improve its decision-making while in the field. While online learning can improve overall performance and adaptability, it also reduces the system's predictability (Defense Innovation Board 2019), so Deployers may not find it suitable for certain missions. Some specific instructions could also be set to minimise unintended results and reduce risk. For example, (Schuller 2017) considers that the versatile command '*whatever you do, do not do X*' could be useful in certain circumstances, and several authors (Sassoli 2014; Thurnher 2012) have proposed a 'shoot last' command where the AWS can be asked to hold fire until fired upon itself.

Of course, the possibility to manipulate any parameters is dependent on design choices made during development. If the system's design does not allow for setting a certain parameter(s) which would have been necessary for legal use, this should have become clear to the Deployer through [$AW_T$]. In such a situation, the Deployer must simply employ Weaponeering to exclude the system from the viable candidates during [*WP*].

**[$CC_O$] *Operational Control*** refers to limiting the Actions the system can inflict upon the Environment (and thereby its Impact) by regulating the scope of the mission or the operational context. Operational Control is most often expressed in terms of spatial or temporal control (Boothby 2018; ICRC 2018; Schmitt 2015; UNIDIR 2014). *Spatial control* refers to restricting the system's reach to a certain geographical area (UNIDIR 2014). To achieve this, the system can for instance be deployed in a stationary manner or be assigned coordinates which bound that system's area-of-operations (Ekelhof 2016; Schuller 2017). Spatial control can help limit the system's operation to only those environments which correspond with its abilities and limitations previously identified in [*AW*], e.g., to only an area with clear ISR, which is uncluttered, or which does not feature civilian concentrations (Roorda 2015). *Temporal control* limits the time in which the system can operate (UNIDIR 2014). The active time of a system can be measured in terms of the period in which it operates independently (e.g., how long a sentry drone is active) or the time-frame between the moment of deployment until force execution (e.g., a loitering munition's air time). Naturally, the probability of failures or unintended consequences increases proportionally with the system's active time (Thurnher 2014), and the Deployer needs to carefully manage

this risk vis-à-vis the expected benefits of employing a longer active time. Temporal control can also account for changes in the environment over time: for example, an area may only be free of civilians for a certain period (Schmitt and Thurnher 2013). Finally, Operational Control can also be achieved by manipulating the operating environment itself. Unintended or unforeseeable interactions can be minimised by implementing measures that prevent changes to the status quo, for example by establishing a perimeter or closing off a section of airspace or terrain to civilian aircraft or vehicles (Boddens Hosang 2021; Roorda 2015).

Finally, *[CC$_L$] Linked Control* is a special form of Context Control that requires particular attention. It is the most direct form of control characterised by the maintenance of a persistent link between the deployed system and a human acting as a pilot/driver or supervisor. This may be the Deployer themselves or someone who has been delegated the task. Either role is represented in Figure 2 as the *Operator*. The Operator interacts with the system through a downlink and an uplink, which require a persistent connection to maintain. A downlink refers to the system sending data, e.g., sensory data or requests for authorisation, 'down' to the Operator, while an uplink refers to the Operator sending data, such as instructions, to the system (Boothby 2019). As the MHC debate does not strictly concern remote-controlled platforms (although the mechanism of sending and receiving data is functionally identical), the arrows representing the downlink and uplink are labelled *supervision* and *intervention* in Figure 2, which better reflects the supervisory role the Operator plays in more autonomous systems.

Linked Control is often framed as the 'ability to intervene' or the requirement of 'positive action by a human operator' for initiating attacks (Article 36 2013a; ICRC 2017). (van den Boogaard and Roorda 2021) call it 'positive control' referring to the "capacity to abort or redirect a system after deployment", as opposed to the term 'procedural control' used to describe [CC$_S$] and [CC$_O$]. Intervention is framed in a number of ways in sources. (UNIDIR 2014) refers to the ability to "intervene, exercise judgment, override or terminate an attack" while (Boardman and Butcher 2019) formulate it as the ability "to impact on the behaviour of the system in time to prevent an undesirable act". Always maintaining the possibility for intervention improves control by allowing the Operator to respond to changing dynamics (Chengeta 2016) or to safeguard against malfunction as part of risk treatment (Stürchler and Siegrist 2017). The ability to abort is emphasised strongly in sources as an important facet of MHC (Boothby 2021; Garcia 2014; ICRC 2016a).

(Roff and Moyes 2016) describe supervision as the ability to "interrogate the system to inform the user or operator about the decisions, goals, subgoals or reasoning that the system used in performing its actions". The supervision component of Linked Control (the downlink) is therefore primarily a prerequisite requirement for enabling intervention, as without knowledge of the system's current status or problems, no proper action can be taken by the Operator. Several authors (Chengeta 2016; ICRC 2018; UNIDIR 2014) also emphasise that that Linked Control should not only be a formality: the Operator must have the interface, information and time necessary to make an informed decision for the control to be meaningful.

One complicating detail concerns whether this facet [CC$_L$] is absolutely required for MHC. The constructive methodology used for this framework, whereby focus was placed on similarities between different positions, functioned well for all previous (sub-)facets, as even when sources would not advance a certain (sub-)facet, there was generally no explicit resistance against adopting it. This is different for [CC$_L$], whereby some sources (IPRAW 2019b; Santoni de Sio and van den Hoven 2018; Schuller 2017; van den Boogaard and Roorda 2021) are actively objecting against the notion that Linked Control is a strict legal requirement. Some take an opposite, similarly strong position (ICRC 2019b; Schuller 2017), namely that it is fundamentally necessary that "humans must participate in decision-making in real time" (Chengeta 2016). Others (Article 36 2017) take a more neutral position, leaving the question open of whether Linked Control is an integral component of MHC. Both sides raise many valid arguments, and this is a complicated legal question which is impossible to address adequately in this paper due to its scope. Therefore, it was decided to

incorporate Linked Control into the framework as $[CC_L]$, but with the caveat that opinions may differ as to whether it is a valid sub-facet.

### 5.4. [PR] Predictability

Predictability is often cited as necessary for a Deployer to exert meaningful control, and appeared before in both lists in Table 1. (UNIDIR 2014) asserts that for "a weapon system to be under control it needs to behave in predictable ways in the environment in which it is deployed and the effect that is intended". (Garcia 2014) draws the connection with the Deployer having to be able to deliberate on "likely incidental and possible accidental effects of the attack" to be in control. How predictability links to control, however, is not commonly elaborated upon. (Article 36 2016) makes a strong but ultimately inadequate attempt, arguing that it "provides a link between commander's intent and the likelihood of outcomes that match that intent". It is likely for this reason that in her own typology of MHC facets, Eklund made the astute observation that predictability seemed to express "a more overarching "goal-element" which the other key elements strive to achieve" (Eklund 2020). The current author ascribes to Eklund's position. Its position as a 'goal-element' can however be very effectively represented in our current framework thanks to the availability of processes. Thus, we will return to this in Section 6.

### 5.5. [AC] Accountability

The so-called accountability gap (Szpak 2020) has been one of the most dominant talking-points in the international debate, and it is unsurprising that it is also frequently proposed as a facet. MHC, according to (Article 36 2016), "requires structures of accountability", a position shared by many others (Geiß and Lahmann 2017; ICRC 2016c; Moyes 2016; Roff and Moyes 2016). (Santoni de Sio and van den Hoven 2018) place the crux of MHC on this very facet, expressing that ultimately, control "concerns responsibility for the consequences of their deployment. How can unacceptable risks be avoided, and how can human beings still be held responsible if systems are acting on their own accord?" (IPRAW 2019a) frames control as requiring a clear delineation of responsibilities: What is actually demanded from the Deployer? As with $[PR]$, however, $[AC]$ is also more of a 'goal-element' than a facet in itself: accountability—as long as we require it to be just[12]—is the result of prior processes enabling the equitable attribution of blame on a person. Similarly to $[PR]$, therefore, we return to this in Section 6.

### 5.6. [$AW_A$] Assessment Awareness

The facet $[AW_A]$ is discussed here at the very end, despite it technically constituting an element of Awareness, due to its timing in the military decision-cycle. Assessment Awareness is rarely mentioned in sources but is added to the framework based on logical inference. The targeting process—or, in fact, any good decision-making process (Boer and van Engers 2013)—places large emphasis on post-execution assessment as part of the feedback loop to determine whether objectives have been fulfilled, problems require addressing, and lessons can be learnt (Department of the Army 2019; North Atlantic Treaty Organisation 2016). Any new information can then be fed back into the pre-deployment analysis to enable continuous improvement (North Atlantic Treaty Organisation 2019). (van den Boogaard and Roorda 2021) do draw the link between assessment and control: "After use, control may be exercised by virtue of a capacity to understand and assess the causes of malfunctions or undesired results from (normal) operation and to take appropriate action (technical and accountability) to prevent similar mistakes." $[AW_A]$ completes the old targeting process and starts a new one, this process repeating until the end of military operations.

---

[12] Of course, the choice could always be made to simply assign accountability to a person—e.g., the Deployer—even in the absence of any knowledge or intent, for example through strict liability. However, many have justifiably argued that this would be fundamentally unfair and contrary to common principles of law (Brazil 2019).



## 6. Processes

Processes link the facets we have explored in Section 5 together, transforming the previously static nodes into steps which impact and influence one another. As an introduction to the concept, one process was already alluded to during our discussion on Interaction Awareness: $[AW_T{\rightarrow}AW_I]$ (see the corresponding arrow in Figure 2). It is evident that in order to understand how a system will operate and react in the operational environment, one must have a reasonable understanding of the system itself (Roff 2016).

### 6.1. [AW→WP,CC] Awareness Informs Weaponeering and Context Control

$[AW{\rightarrow}WP,CC]$ is a more substantial process but very intuitive. For a commander to make a proper weaponeering decision, they "must be fully informed as to [their] tactical options, as well as the anticipated consequences of implementing these precautions" (Corn 2014). Thus, a solid understanding of all three Awareness facets is required: $[AW_T{\rightarrow}WP]$ the weapon's abilities, parameters and limitations, $[AW_O{\rightarrow}WP]$ the situation on the ground, based on collected ISR, and $[AW_I{\rightarrow}WP]$ how it will perform concretely in these conditions (Curtis E. Lemay Center 2019; UNIDIR 2014). Similarly, specific Context Control actions also rely on the information provided by $[AW]$. (United Kingdom 2020) confirmed that "[c]ontext matters when considering appropriate control measures. The nature of the task and the environment should have major implications for how control is implemented". A particular example by (Roorda 2015) illustrates how $[AW]$ directly impacts $[CC]$, even in negated form. He postulates that if a commander has great difficulty with $[AW_T]$ (e.g., it uses an extremely advanced or non-transparent AI that is difficult to comprehend), they would be more "inclined to restrict its use due to a lack of understanding of how the system would respond to circumstances (predictable or not) and what effects it would generate". In other words, deficiencies in $[AW]$ can be compensated by greatly intensifying $[CC]$.

### 6.2. [AW,CC→PR] Awareness and Context Control Permit Prediction

$[AW{\rightarrow}PR]$ is perhaps the process which is mentioned the most often. It is also nestled in Roorda's quotation above: a deficient $[AW]$ leads to a lack of ability to predict the system's responses. (Moyes 2016) argues that predictability is fundamentally a characteristic of the interaction between the system and environment, and thus, from "an understanding of the technology and the context in which it will operate, a commander should be able to assess likely outcomes, including the risk of civilian harm, which is the basis for the legal assessment". A clear example of $[AW_T{\rightarrow}PR]$ is non-transparent AI: the less understandable a model, the more difficult it is to predict the consequences of use (Goussac 2019). (ICRC 2019b) added that predictability "will depend not only on the technical design of the system, but on variations in the environment over time and the interaction of the system with that environment, taking into account the task it is used for", clearly referencing $[AW_C,AW_I{\rightarrow}PR]$.

Prediction is also generated by Context Control (Moyes 2016; Schuller 2017; UNIDIR 2014). $[CC{\rightarrow}PR]$ is implicit in the fact that the goal of $[CC]$ is to impose restrictions on the system in order to improve reliability and reduce risk. (Crootof 2015), for example, argues that predictability is a direct function of the active time of a system: the more prolonged, the higher the probability that intervening, unforeseen factors may introduce themselves into the environment. (Stürchler and Siegrist 2017) reference $[CC_S,CC_O{\rightarrow}PR]$ when they comment that predictability "can also be increased by restricting the autonomous weapon systems' parameters of engagement in line with the system's compliance capabilities". As such, this confirms our and Eklund's earlier assumption that Predictability is more of a 'goal-facet': its fulfilment is a function of $[AW]$ and $[CC]$.

### 6.3. [AW,WP,CC,PR→AC] Processes Leading to Accountability

Finally, let us consider processes leading into $[AC]$, Accountability. Clearly delineating the processes here is particularly important as accountability generally requires specific conditions to be met for the attribution of responsibility to be equitable (whether it be

moral or legal accountability). Indeed, the apparent difficulties of achieving this when an autonomous system functionally made the problematic decision lie at the core of the accountability gap argument (Anderson and Waxman 2013; Asaro 2012; Human Rights Watch 2015; Sparrow 2007). There are generally two main components to accountability: a mental component and causality.[13] The accountability gap argument asserts that no single human will possess both elements at once, leading to an inability to assign accountability—an obviously undesirable result. For instance, the programmer may be knowledgeable, but their involvement is likely to be deemed too distant from the actual decision moment (ICRC 2014). The commander is more proximate temporally and causally, but the cognitive and volitive elements may be lacking because they may claim to have never foreseen the malfunction.

Our new MHC framework provides an elegant solution to this dilemma, in the form of processes feeding into $[AC]$. First, $[AW{\rightarrow}AC]$ and $[PR{\rightarrow}AC]$ contribute toward the mental element. (United Kingdom 2020) underlined this quite well when it remarked that holding commanders responsible would necessitate that they "have a sufficient understanding of the capabilities and limitations of the weapon system, and of the environment in which it is to be deployed". (ICRC 2018) also argued that accountability can be ensured through, *inter alia*, an understanding of the system, the environment and how the system interacts with it. Awareness and Predictability connect the Deployer's intent and knowledge with the eventual outcome (ICRC 2018), and eliminate many possible excuses a Deployer could raise to claim ignorance or an inability to know. For example, this following argument, presented as a conundrum and a challenge to AWS use, would no longer be convincing in the presence of $[AW,PR{\rightarrow}AC]$ processes:

> [T]he limits of control over, or the unpredictability of, an autonomous weapon system could make it difficult to find individuals involved in the programming and deployment of the weapon liable for serious violations of IHL. They may not have the knowledge or intent required for such a finding, owing to the fact that the machine can select and attack targets independently. (ICRC 2016b)

The $[WP{\rightarrow}AC]$ and $[CC{\rightarrow}AC]$ processes cover the causal element of responsibility. Here, the common accountability gap argument would sound as follows: In the absence of $[CC_L]$ (i.e., there was no Linked Control and the machine did indeed make the final decision independently),[14] there is no proximate act of the Deployer which we can connect with the result. Take the example of an autonomous car. Some would claim that these vehicles "make driving decisions without any human intervention, and therefore no human is directly responsible in the case of an accident" (Righetti 2016). However, $[WP,CC{\rightarrow}AC]$ clearly shows there is. As we have seen previously, $[WP]$ is the most involved form of control that can be exercised. If, after being aware of all the components of $[AW]$, the Deployer still makes a reckless decision and deploys a weapon that is badly paired with the intended operational context, or fails to enact the proper Context Controls that would be expected of a reasonable officer in his position (Additional Protocol I 1977), they have clearly 'caused' the violation to occur.

If Prediction was described by Eklund as an 'end-goal' of $[AW]$ and $[CC]$, Accountability is the end result of the entire MHC scheme. If one has properly implemented all preceding facets, accountability follows naturally. $[AW]$ and $[PR]$ ensure that the Deployer has the knowledge and foresight necessary for accountability, while $[WP]$ and $[CC]$ constitute the actual (wrongful) acts which will have caused the result.

---

[13] This is generally true for both legal and moral accountability, even though the exact scope of these elements may vary. See e.g., (Additional Protocol I 1977) (IHL); (Rome Statute 1998) (criminal law); (Marauhn 2018; Sparrow 2007) (moral/philosophical).

[14] There generally is no problem of accountability for systems which apply $[CC_L]$, as the Operator (the system pilot or supervisor) would be responsible for (and directly cause) the system's final decision.

## 7. Concluding Remarks

Having combined and integrated all proposals for MHC criteria, a workable foundation for addressing many concerns related to the use of AWS has emerged. While the framework presented in this paper is only the current author's synthesis of the various positions in the discourse, it nevertheless is a significant step forward to rectifying MHC's most significant flaw as a legal or policymaking concept: its lack of a unifying theory. A particular added value of this framework is the elaboration of a more concrete list of actions which commanders can take to exercise meaningful control over their AWS, and how these controls impact each other. This added clarity can guide further legal analysis, policymaking and negotiations by providing a common baseline which is anchored in—and thus compatible with—actual targeting practice. It also can be used as a toolbox to address common concerns related to AWS. We have seen one example of this during our discussion on the popular accountability gap argument (Section 6.3) and how this can be addressed through aforementioned controls, fulfilling the volition and causation requirements through prior expressions of MHC.

A framework is only the first step in properly developing MHC as a guiding principle or legal requirement for AWS. The next step is to test it against a certain legal regime such as IHL to determine the precise requirements encapsulated in each facet. How much transparency should States design into their systems to enable $[AW_T]$? How much ISR is demanded by IHL to fulfil $[AW_C]$? What are the legal principles that determine a weaponeering decision? What are the particularities of legal and criminal accountability and where are the obstacles in translating $[AW]$, $[CC]$ and $[PR]$ into such accountability? Such details are crucial for moving the framework beyond the theoretical and applying it in practice. The author hopes to return to these questions in a future paper.

There will be individual disagreements on the details. A major one which we have touched upon relates to $[CC_L]$, Linked Control. Is this always required, and how should it be defined specifically? Is there a legal basis for such an assertion, or is it purely made on ethical or moral terms? The author strongly recommends further discussion on such matters, adjusting and refining our understanding of MHC to eventually reach a consensus.

This framework placed the Deployer as the figure ultimately responsible for ensuring MHC. While most sources tend to agree that the Deployer (or 'commander', as it is frequently operationalised) should take primary responsibilities for any consequences of weapon use (State of Israel 2019), our discussion has revealed that the fulfilment of many MHC facets is also dependent on prior decisions, be it during design, development, adoption, or at the strategic level of the campaign. Thus, some have also suggested that "accountability must also come to bear upon wider systems or organizations that produce such socio-technical systems and artifacts" (Roff and Moyes 2016). While this broader form of accountability was not the focus of the current framework, it is nonetheless important to consider how actors beyond the targeting cycle also contribute to, and to what extent they can be held accountable for, MHC in AWS.

Finally, while the main aim of this paper was to develop the notion of meaningful human control, its findings are not bound by the term 'MHC' itself. The conclusions of this framework are generally applicable to any debate on AWS as its arguments are not based on MHC as a philosophical or semantic concept: rather, the framework was constructed based on legal norms in force and actual military practice, particularly the targeting cycle. As such, even if the term 'MHC' is hypothetically abandoned tomorrow by the international community for a different related concept, the ideas and recommendations proposed in this paper remain applicable. It is the core messages behind the facets and processes that should be retained, and not the term MHC per se.

**Funding:** This research received no external funding.

**Institutional Review Board Statement:** Not applicable.

**Informed Consent Statement:** Not applicable.

**Data Availability Statement:** Not applicable.

**Conflicts of Interest:** The author declares no conflict of interest.

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
