# Peer review of "A Practicable Operationalisation of Meaningful Human Control"

_laws_

Round 1
Reviewer 1 Report
This is an excellent article providing for a useful and partly innovative framework for the analysis what meaningful human control means. The author masters scholarly writings and positions of States and international organizations. An important proposition is that predictabbility and accountability are not facets of meaningful human control, but goals. The author answers all objections I had when reading some of his statements. He or she carves out the relatively few real substantive controversies - on which the author is, however, very reluctant to express an opinion. The only criticism i could think of is that the author possibly underestimates the inherent "black box" aspect of artificial intelligence in his very convincing developments on the "awareness" facet.
The form and references are excellent, although the alphanumerical cross-refences (e.g. 1a or 1o) make the reading sometimes challenging (but they are always correct).
Reviewer 2 Report
The main problem is that author make just summary of the propositions concerning MHC; and the only novelty lies in the synthesis. It is telling that in the introduction there is no thesis, author promises just summary and synthesis of the propositions already expressed in the literature. Therefore I would really emphasise in the introduction what is the added value of the research of author.
In addition figure 2 which should be crucial for this research is messy, it really does not help to understand the problem, Author needs to rethink the whole graphic as now it complicates instead of helping understand the problem.
Bibliography is a mess, it is not ordered in alphabetical order so it is difficult to find particular source (of course in printed version)
